# Evolution of the HIV-1 integration site landscape and inducible reservoir in early-treated people

Tine Struyve[1], Marion Pardons[1], Jozefien De Clercq[1], Liesbet Termote[1], Laurens Lambrechts[1], Ytse Noppe[1], Mathias Lichterfeld[2,3], Sofie Rutsaert[1☯], Linos Vandekerckhove[1☯*]

**1** HIV Cure Research Center, Department of Internal Medicine and Pediatrics, Ghent University Hospital, Ghent University, Ghent, Belgium, **2** Infectious Disease Division, Brigham and Women's Hospital, Boston, Massachusetts, United States of America, **3** Ragon Institute of MGH, MIT and Harvard, Cambridge, Massachusetts, United States of America

☯ These authors contributed equally to this work.
* Linos.Vandekerckhove@UGent.be

## Abstract

Persistence of the HIV-1 reservoir is the major barrier to a cure. Little is known about the dynamics of the proviral integration site landscape and inducibility of the viral reservoir in early-treated individuals. Here, we perform a longitudinal analysis of the viral reservoir in individuals who started treatment during acute infection and compare these findings to chronically-treated individuals. Even in early-treated individuals, clonal expansion contributes to reservoir persistence. Integration site analysis reveals similar distributions after one year of antiretroviral therapy (ART), irrespective of treatment initiation timing. Notably, proviruses integrated in heterochromatin regions are already detected in early-treated individuals after one year on ART and are progressively enriched after five years on ART, suggesting post-integration selection mechanisms. Using a lipid nanoparticle containing Tat mRNA (Tat-LNP) in combination with phorbol myristate acetate (PMA), we detect for the first time the inducible reservoir in individuals treated during acute infection with small reservoir sizes. Furthermore, we show that, in both the acute and chronic cohorts, the inducible reservoir shifts towards a more differentiated T cell compartment over time. Collectively, these findings indicate that clonal expansion and integration site selection contribute to reservoir persistence early after ART initiation in individuals treated shortly after seroconversion.

## Author summary

Despite ART, HIV persists in a small pool of infected cells. This reservoir is the main barrier to curing HIV. We studied how this reservoir changes over time in people who started treatment very early after infection and compared them to

**Data availability statement:** All relevant data are within the manuscript and its Supporting information files. HIV-1 sequence data that support the findings of this study have been deposited in GenBank with the accession codes: PQ286693 - PQ286864 and PQ286865 - PQ286941. Scripts concerning de novo assembly of HIV-1 genomes can be found at the following GitHub page: https://github.com/lau-lambr/virus_assembly. Scripts concerning the integration site feature analyses can be found at the following GitHub page: https://github.com/HCRCugent/Integration_site_feature_analysis.

**Funding:** This research was supported by multiple funding sources. From FWO Vlaanderen, support was provided through FWO JUNIOR (G0B3820N), FWO SENIOR (G0A1824N and G0ABH25N), the SBO-SAPHIR project (S000319N) and 1.8.020.09.N.00 to LV. Individual fellowships by FWO Vlaanderen included support for T.S. (1SA7125N) and L.L (1S29220N). M.P. received postdoctoral funding from VLAIO O&O (HBC.2018.2278). J.DC. is funded by Ghent University Special Research Fund (BOF) doctoral scholarship (BOF19/DOC/220). Additional support was provided by the National Institutes of Health (NIH), including grants R01 AI152979-01 (MPI: L.V. and M.L.), NIH MDC grant (RID-HIV: UM1AI164561). Research in the lab of M.L. is supported by NIH grants AI184094, AI152979, AI155233, AI135940, AI176579. M.L. is a member of the DARE, ERASE, PAVE and BEAT-HIV Martin Delaney Collaboratories (UM1 AI164560, AI164562, AI164566, AI164570). The content is solely the responsibility of the authors and does not necessarily represent the official views of the National Institutes of Health. The authors declare no conflicting financial interests. The funders had no role in study design, data collection and analysis, decision to publish, or preparation of the manuscript.

**Competing interests:** The authors have declared that no competing interests exist.

people who started later. We found that, even with early treatment, the infected cells persist and expand over time. After 5 years on ART, in people who started treatment early, HIV becomes enriched in regions of the genome that are less transcriptionally active, suggesting that selective pressures shape which infected cells persist. We also applied a new combination of latency reversal agents, allowing us for the first time to detect the inducible reservoir even in people with very small reservoir sizes. Finally, we observed that the inducible reservoir shifts towards more differentiated types of immune cells over time on ART. Together, our findings provide new insights into how HIV persists despite early treatment.

## Introduction

Despite antiretroviral therapy (ART), HIV persists in a pool of long-lived memory CD4 T cells, referred to as the viral reservoir [1–3], which is the major barrier to a cure. During the past decade, multiple studies have provided evidence that early initiation of ART within 100 days of infection improves health outcomes and reduces risk of HIV transmission [4–6]. However, although early initiation of ART limits the seeding of the viral reservoir [7–11] and genetic diversity [12–15], it does not prevent viral rebound after treatment interruption [16,17]. Therefore, it is critical to better understand the composition of the viral reservoir on ART and the mechanisms contributing to its persistence in early-treated individuals.

Previous studies have characterized the integration site landscape in early-treated pediatric cohorts [18,19]. However, the dynamics of clonal expansion and the integration site landscape in early-treated adults remain largely uncharacterized [20,21]. Indeed, little is known about the longitudinal contribution of clonal expansion to HIV-1 persistence during ART in early-treated adults. Moreover, a growing body of evidence indicates that the integration site of proviruses plays a role in the persistence and potential reactivation of the virus [22–27]. Positive selection of proviruses integrated into repressive chromatin regions, including centromeric satellite DNA and Krüppel-associated box zinc-finger protein (KRAB-ZNF) genes on chromosome 19, has been observed in both elite controllers [22] and individuals on long-term ART (~20 years) [26]. Centromeres are heterochromatin domains composed of repetitive, gene-poor DNA and organized into tightly packed nucleosomes that limit accessibility to the transcriptional machinery [28,29]. Moreover, the human heterochromatin protein CBX1 coats large KRAB-ZNF gene clusters on chromosome 19 [30], establishing a repressive chromatin environment. Nevertheless, this selective integration site pattern has not yet been investigated in early-treated individuals over a longitudinal time course.

In addition, an in-depth assessment of the inducible reservoir in individuals who initiated ART during acute infection is still lacking. In one study, none of the six participants displayed detectable levels of translation-competent proviruses after phorbol myristate acetate (PMA)/ionomycin stimulation [21], presumably due to the small reservoir size associated with early treatment initiation. Therefore, potent latency reversal agents (LRA) are required to enhance reactivation above the current limit

of detection. In our recent study, stimulation with an HIV-1 Tat mRNA-containing nanoparticle (Tat-LNP) resulted in higher frequencies of p24+cells compared to the gold standard PMA/ionomycin [31], offering new opportunities to detect and characterize the translation-competent reservoir in early-treated individuals.

Taken together, only a limited number of studies have investigated the viral reservoir after early treatment initiation and mainly lack: (i) longitudinal timepoints on ART, to assess the impact of ART duration on the persistence of the viral reservoir and (ii) proper reference cohorts with similar treatment durations but initiation of ART during chronic infection, to assess the effect of treatment initiation timing. While early ART initiation is known to limit the size of the reservoir, its impact on qualitative features, such as proviral integration site distribution and inducibility, remains unclear. Here, we performed an in-depth longitudinal analysis of the viral reservoir in individuals who started treatment during acute infection. In addition, we included reference cohorts with similar treatment durations but initiation of ART during chronic infection to assess the impact of treatment initiation timing. We characterized the HIV-1 integration site landscape to (i) compare integration site features between the acute and chronic cohorts and (ii) study selection mechanisms over time in early-treated individuals. Moreover, using a combination of Tat-LNP and PMA to allow for maximal in vitro reactivation, we studied the translation-competent reservoir by simultaneously investigating the inducibility, along with the integration site and proviral genome of p24-expressing cells. Using this approach, we (i) compared the inducibility in the acute and chronic cohorts and (ii) studied how the composition of the inducible reservoir evolves over time.

## Results

### Clinical and virological characteristics of the participants

We performed an in-depth characterization of the HIV-1 viral reservoir in nine individuals who initiated ART during acute infection (Fiebig II-III: n=5; Fiebig IV-V: n=4; Fig 1A and S1 Table). Peripheral blood mononuclear cells (PBMCs) from leukapheresis were collected after a median of 0.72 years on suppressive ART (range 0.31-1.93 years; referred as Acute undetectable (UD)). Additionally, matched longitudinal blood samples of five participants were collected after a median of 5.33 years on ART (range 3.41-7.42 years; referred as Acute UD+5; Fig 1A and S1 Table). For comparative purposes, two reference cross-sectional cohorts of individuals who initiated ART during chronic infection were included in this analysis: one cohort with blood sampling at UD (n=10; range 0.2-1.3 years on ART; referred as Chronic UD; Fig 1B) and one cohort with blood sampling at UD+x (n=57; range 1.4-24.8 years on ART; referred as Chronic UD+x; Fig 1C and S1 Table). As longitudinal samples were not available for the Chronic UD individuals, we included the cross-sectional Chronic UD+x cohort, which comprises samples collected over a range of time on ART. All participants maintained an undetectable viral load (<20 copies/mL) at the time of sampling.

The frequency of infected cells was measured with the Rainbow assay [32], a digital PCR method that simultaneously measures total and intact HIV DNA (S1A–E Fig). At the UD timepoint, the acute cohort displayed a significantly lower frequency of cells with total HIV-1 DNA in comparison with the chronic cohort (medians=290 and 2398 copies/$10^6$ CD4 T cells in Acute UD and Chronic UD, respectively; p=0.0002; S1A Fig). This trend was also observed at the longitudinal timepoint (medians=108 and 555 copies/$10^6$ CD4 T cells in Acute UD+5 and Chronic UD+x, respectively; p=0.04). All participants in the acute cohort displayed a decrease in the frequency of cells harboring total and intact HIV-1 DNA from UD to UD+5 (p=0.06 for both total and intact HIV-1 DNA; S1B Fig). In addition, we evaluated the genetic viral diversity among individuals for all cohorts and timepoints (S1F–I Fig). At UD the average intra-individual genetic diversity was lower in the acute cohort compared to the chronic cohort (medians=1.7 bp and 46.1 bp in Acute UD and Chronic UD, respectively; p=0.07; S1F Fig). No further sequence diversification was observed during prolonged suppressive ART in the acute (p=0.13; S1G, H Fig) cohort.

### Clones appear after increased time on ART in early-treated individuals

Using Integration site loop amplification (ISLA), we profiled the integration site landscape of proviruses in the acute (UD and UD+5) and chronic (UD) cohorts. In total, we obtained 452, 124 and 197 integration sites for the Acute UD (n=8),

PLOS Pathogens

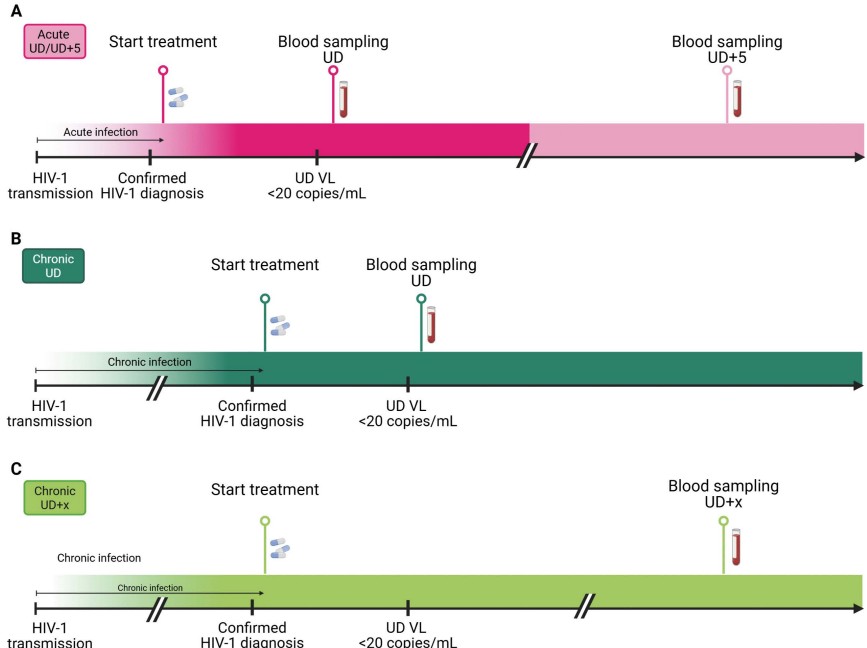

**Fig 1. Schematic overview of participant sampling timeline.** (A) Individuals with a confirmed HIV-1 diagnosis started treatment during acute infection and were followed up in time. Blood sampling was performed at undetectable (UD; n = 9)) viral load (< 20 copies/mL) and again after 5 subsequent years (UD + 5; n = 5). (B) Individuals with a confirmed HIV-1 diagnosis who began treatment during chronic infection, with blood sampling at undetectable viral load (UD; n = 10). (C) Individuals with a confirmed HIV-1 diagnosis who began treatment during chronic infection, with blood samples collected at varying durations on ART (UD + x, n = 57). Created in BioRender. Struyve, T. (2025) https://BioRender.com/cpljl4y.

Acute UD + 5 (n = 5) and Chronic UD (n = 3) cohorts, respectively (S2 Table). We defined clonal expansion as the recurrence of at least two identical integration sites. At UD, in Fiebig group II-III, no integration sites (n = 214) were detected more than once in any of the four participants (Fig 2A). In Fiebig group IV-V, three clonally expanded populations of infected cells (integration in *GPN1:* 9 recurrences, integration in *CBFB* and *ASB16-AS1*: 2 recurrences) were detected in two out of four individuals and accounted for 5% of total integration sites (13/238). In contrast, all three individuals from the Chronic UD cohort displayed multiple distinct integration sites detected more than once. Therefore, the overall percentage of clonally expanded cells was higher in the chronic cohort compared to the acute cohort (Acute UD: 2.88%; Chronic UD: 7.61%; Fig 2B). Longitudinal analysis in the acute cohort showed an increase in integration sites detected more than once at UD + 5 compared to UD (Acute UD: 2.88%; Acute UD + 5: 12.9%; Fig 2B), with clonally expanded cells detected in three out of the five individuals. To provide a quantitative measure of clonal expansion, we calculated the oligoclonality index (OCI) [33] for each individual (Fig 2C). Values range between 0 and 1, where 0 indicates complete heterogeneity (each integration site is detected only once), and 1 indicates complete homogeneity (all integration sites come from a single clone). At UD, the OCI was higher in the chronic cohort compared to the acute cohort (medians = 0.029 and 0.061 in Acute UD and Chronic UD, respectively; p = 0.28). Within the acute cohort, the OCI increased from UD to UD + 5 among the five longitudinally followed individuals (paired analysis: medians = 0.028 and 0.082 in Acute UD and Acute UD + 5, respectively; p = 0.0625); suggesting an increase in clonal expansion. Interestingly, at UD + 5, participant PA05 (Fiebig group II-III) harbored four clonal populations which accounted for half of the total integration sites detected in that participant (Fig 2A). Furthermore, we found one shared integration site between both timepoints in only one participant (PA15). Collectively, these longitudinal observations underscore that the reservoir persists through clonal proliferation, even when treatment is initiated during acute infection.

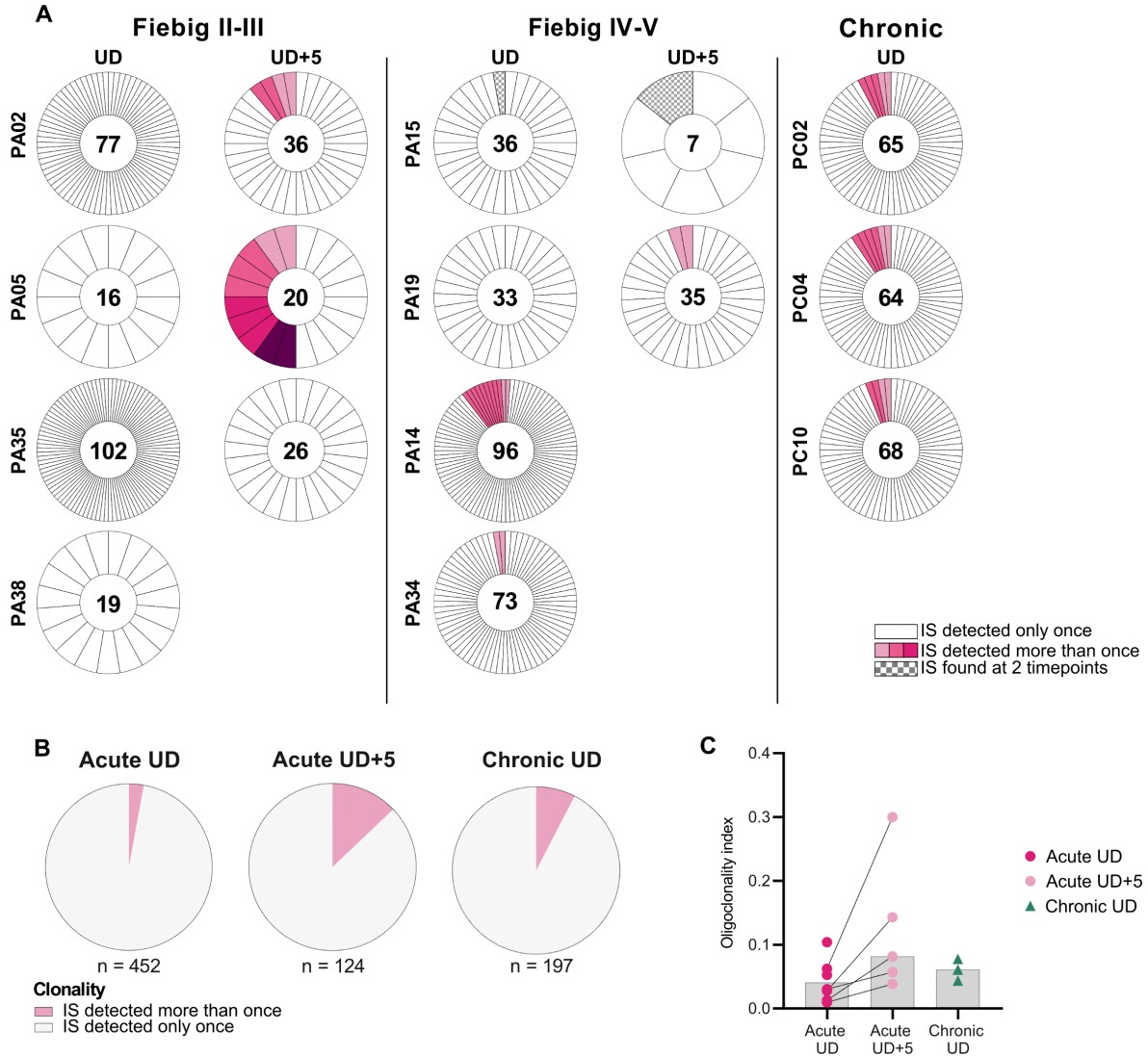

**Fig 2. Clonal expansion dynamics on ART in individuals treated during acute and chronic infection.** (A) Individuals are categorized into three groups according to ART initiation timing [during acute infection: Fiebig II-III (left panel), Fiebig IV-V (central panel); during chronic infection (right panel)]. Donut charts displaying the number of integration sites (IS) detected only once (white) or more than once (colored) at UD and UD+5. Distinct clonal populations of infected cells are represented by different shades of pink. The total number of IS generated by ISLA for each participant is shown in the middle of each donut. (B) Pie charts displaying proportions of integration sites (IS) detected once (white) and more than once (pink). (C) Oligoclonality index for each individual of the Acute UD (n=8), Acute UD+5 (n=5) and Chronic UD cohort (n=3). Grey bars depict median values. Lines connect paired values for the longitudinally followed individuals in the acute cohort.

## Progressive selection of proviruses in repressive chromatin regions over time on ART in early-treated people

We then compared the integration site landscape between timepoints and cohorts. Integrations were more frequent in opposite orientation to host genes (Fig 3A) and in genic regions (Fig 3B) among all cohorts. Notably, proviral integration in centromeric/satellite DNA (0.88% of total IS at UD) and/or KRAB-ZNF genes on chr19 (0.88% of total IS at UD) was already detectable in five of the eight individuals of the Acute UD cohort (Figs 3B and S2). Moreover, the acute and chronic cohorts at UD showed similar levels of integration within centromeric/satellite DNA (Acute UD: 0.88% and Chronic

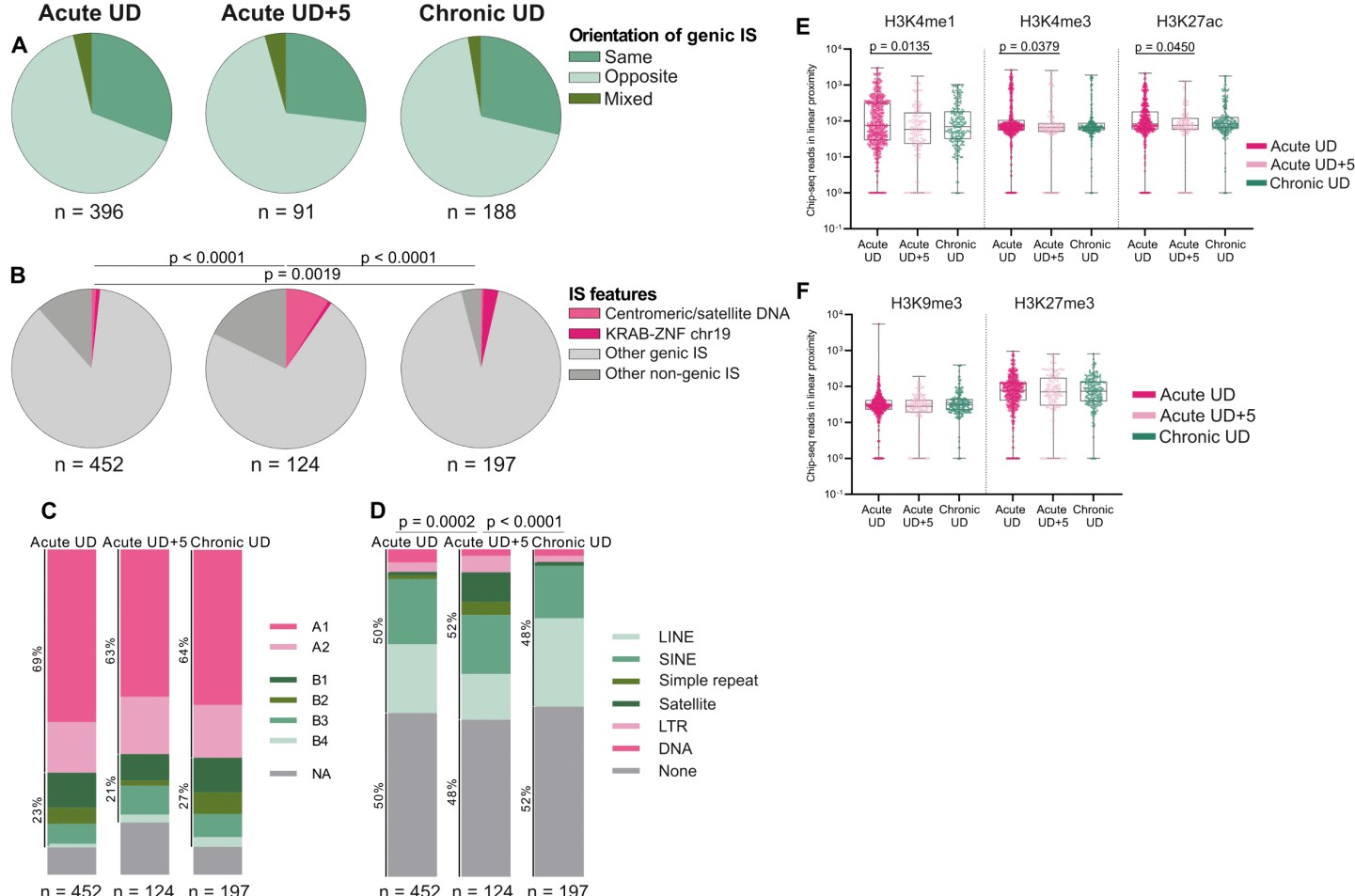

**Fig 3. Integration site features of proviruses from Acute UD, Acute UD+5 and Chronic UD individuals.** (A) Pie charts displaying proportions of IS in genic regions with same, opposite and mixed orientation relative to the host gene. (B) Pie charts displaying proportions of IS in centromeric/satellite DNA, ZNF genes with KRAB domain on chr19, other genic regions and other non-genic regions. (C) Bar plots showing the proportions of IS in Hi-C compartments A1-A2 and B1-B4. Cumulative proportions of IS in compartments A and B are indicated on the left side of the graph. (D) Bar plots showing the proportions of IS in repetitive elements. Cumulative proportions of IS in repetitive elements are indicated on the left side of the graph. (E-F) Boxplots showing ChIP-Seq reads corresponding to activating (E) or inhibiting (F) histone features in linear proximity (+/-5 kb) of IS. (A-D) n represent the number of IS in the analysis. Chi-squared test (A) or Fisher's exact test (B-D) were performed. (E-F) Two-tailed Mann-Whitney tests were performed for Acute UD-Acute UD+5, and Acute UD-Chronic UD.

UD: 0.51%; p > 0.99) and KRAB-ZNF genes on chr19 (Acute UD: 0.88% and Chronic UD: 3.05%; p = 0.07). Interestingly, paired analysis of the five longitudinally followed individuals from the acute cohort revealed an increase in the proportion of integrations in centromeric/satellite DNA with time on ART in three of the five individuals (UD: 1.52%; UD+5: 8.87%; p = 0.001), suggestive of a positive selection of viruses integrated into heterochromatin regions (S2 Fig). Of note, in individual PA35 at UD+5, 19% of the retrieved proviruses were integrated in distinct locations of (peri-) centromeric DNA (S2 Fig). Furthermore, individual PA05 harbored 2 clones in centromeric DNA at UD+5, accounting for 25% of the retrieved integration sites in that participant.

Aligning our IS data to three-dimensional chromosomal interaction data obtained through Hi-C sequencing [34] revealed that proviruses were predominantly positioned within the active chromatin compartments A1 and A2 in all cohorts

(Acute UD: 69%; Acute UD + 5: 63%; Chronic UD: 64% of total integration sites; Fig 3C). Analysis of integration in repetitive elements using RepeatMasker showed that proviruses were preferentially found into long and short interspersed nuclear elements (LINE, SINE) among all cohorts (Acute UD: 41%; Acute UD + 5: 31%; Chronic UD: 43% of total integration sites; Fig 3D). No overall significant differences in repeat patterns were found between Acute UD and Chronic UD (p = 0.58). However, longitudinal analysis of integration sites in the acute cohort revealed an overall alteration of integration in repeat patterns (p = 0.0002), with a marked increase of integration sites in satellite DNA at UD + 5.

To further characterize the chromatin landscape at integration sites, we assessed the enrichment of activating and inhibiting histone marks near proviruses using chromatin immunoprecipitation (ChiP-seq) data from blood-derived primary T cells [35]. We noted no significant difference in the number of ChiP-seq reads corresponding to activating or inhibiting histone marks in proximity to proviruses between the Acute UD and Chronic UD cohorts. However, longitudinal analysis of the acute cohort revealed a decline in proviruses located in proximity to activating histone marks H3K4me1 (p = 0.01), H3K4me3 (p = 0.04) and H3K27ac (p = 0.05) from UD to UD + 5 (Fig 3E). No difference was observed for the number of ChiP-seq reads related to inhibitory histone marks between the two timepoints in the acute cohort (H3K9me3, H3K27me3; Fig 3F).

Together, these findings indicate that similar integration site patterns are observed between both the acute and chronic cohorts at UD. Interestingly, integration into repressive chromatin regions, including centromeric DNA and KRAB-ZNF gene clusters on chr19, is already detectable early after ART initiation in more than half of the participants of the acute cohort (5/8) and significantly increases over time on ART in several individuals (3/5).

## Inducibility of the viral reservoir in acute and chronic cohorts

A comprehensive analysis of the translation-competent reservoir, defined as HIV-infected cells harboring proviruses capable of producing viral proteins, in early-treated individuals under ART remains unexplored. These proviruses may contribute to persistent immune activation and viral rebound if ART is interrupted. We used the HIV-Flow assay [36] to quantify and characterize CD4 T cells expressing the HIV-1 p24 capsid protein following stimulation. We combined Tat-LNP with PMA to obtain maximal reactivation in vitro. This combination of LRAs was 2.5-fold and 9.5-fold more potent than the previously described combinations of LRAs Tat-LNP/panobinostat (PNB) or PMA/ionomycin [31], respectively (S3 Fig).

At 24h post-stimulation, the frequency of p24 + cells for the Acute UD cohort ranged between 0.4 and 20.3 p24 + cells/$10^6$ CD4 T cells (median frequency = 1.4 p24 + cells/$10^6$ CD4 T cells; Figs 4A, S4, S5A, and S6). Although not significant, there was a trend towards a lower frequency of p24 + cells in the acute cohort compared to the chronic cohort at both timepoints (medians = 1.4, 3.2, 4.5, 22.2 p24 + cells/$10^6$ CD4 T cells for Acute UD (n = 9), Acute UD + 5 (n = 5), Chronic UD (n = 10), Chronic UD + x (n = 11), respectively), suggesting a lower inducible reservoir size in early-treated individuals (Fig 4A). However, normalization of the frequency of p24 + cells to total HIV-1 DNA revealed no significant differences between Acute UD (1.39%) and Chronic UD (0.5%; p > 0.99)), or between Acute UD + 5 (2.27%) and Chronic UD + x (1.58%; p > 0.99), suggesting similar proportions of HIV-infected cells with an inducible provirus between both cohorts (Figs 4B and S6D). Moreover, the size of the translation-competent reservoir was maintained in the acute cohort after prolonged ART (Figs 4A and S6C), underscoring its persistent nature despite early treatment initiation.

## Phenotypic differentiation of p24 + cells is observed after time on ART

We first compared the phenotype of the p24- cells to the p24 + cells for each cohort (Fig 4C). In early-treated individuals, PMA/Tat-LNP-stimulated p24 + cells were enriched in the effector memory fraction (TEM) compared to p24- cells, although the limited number of points did not allow to reach statistical significance (Acute UD: p = 0.09; Acute UD + 5: p = 0.38). The same trend was observed for the Chronic UD + x cohort (p = 0.004; Fig 4C), consistent with previous reports using PMA/ionomycin or Tat-LNP/PNB stimulation in chronically-treated individuals [21,36,37]. Furthermore, to determine whether timing of treatment initiation alters the inducible viral reservoir's composition within specific cell subsets, we pooled

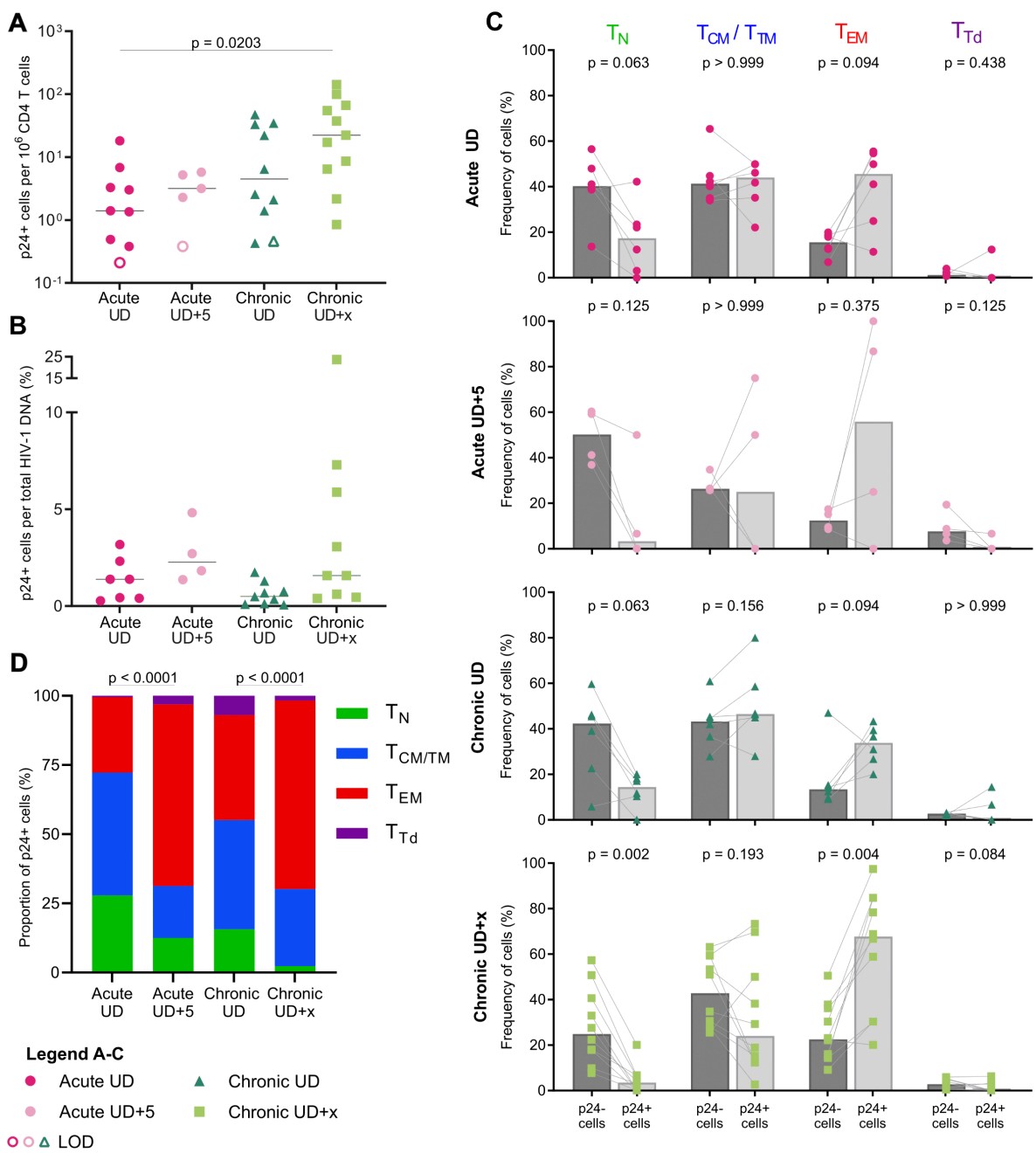

**Fig 4. Frequencies and phenotypes of p24 + cells following PMA/Tat_LNP stimulation.** (A-D) CD4 T cells from Acute UD, Acute UD + 5, Chronic UD and Chronic UD + x individuals were stimulated for 24h with PMA/Tat_LNP (A) Frequencies of p24 + cells in CD4 T cells. Median values are plotted. Open circles represent limit of detection (LOD). (B) Percentages of HIV-infected cells expressing the p24 protein following stimulation. Median values are plotted. Participants with an LOD frequency of p24 + cells in CD4 T cells or unreportable/unavailable total HIV DNA quantification were not included in this analysis. (C) Frequencies of cells with a given phenotype (TN, TCM/TTM, TEM, TTd) in p24+ and p24- fractions. Participants with a minimal number of 4 p24 + cells are represented. Grey bars depict median values. (D) Proportions of all p24 + cells with a given phenotype (TN, TCM/TTM, TEM, TTd). p24 + cells (Acute UD: 187, Acute UD + 5: 32, Chronic UD: 287, Chronic UD + x: 633) from all participants are included. (A-B) Kruskal-Wallis tests were performed (Dunn correction for multiple comparisons). (C) Two-tailed Mann-Whitney tests were performed. (D) Fisher's exact tests were performed for Acute UD-Acute UD + 5, and Chronic UD-Chronic UD + x. TN = naïve T cells, TCM/TTM = central and transitional memory T cells, TEM = effector memory T cells, TTd = terminally differentiated T cells.

all p24 + cells and compared the phenotype of these cells between timepoints (Figs 4D and S5B). When analyzing all p24 + cells, the acute and chronic cohorts at UD displayed a higher fraction of p24 + cells residing in the naïve (TN) subset compared to the UD + 5/UD + x timepoints (Acute UD: 27.8%, Acute UD + 5: 12.5%, p = 0.08; Chronic UD: 15.68%, Chronic UD + x: 2.37%, p < 0.0001; Figs 4D and S5B). The same trend was observed for the proportion of p24 + cells residing in the central memory/transitional memory (TCM/TTM) subsets (Acute UD: 44.39%; Acute UD + 5: 18.75%, p = 0.0063; Chronic UD: 39.37%; Chronic UD + x: 27.8%, p = 0.0006). In contrast, the Acute UD + 5 and Chronic UD + x groups displayed a higher frequency of p24 + cells residing in the TEM fraction compared to the Acute UD and Chronic UD groups (Acute UD: 27.27%, Acute UD + 5: 65.63%, p < 0.0001; Chronic UD: 37.98%, Chronic UD + x: 68.09%, p < 0.0001). These findings suggest that with increased time on ART, there is a shift in the subset composition of the translation-competent reservoir towards more differentiated cellular phenotypes.

## Proviruses in centromeric regions can be reactivated from latency following PMA/Tat-LNP stimulation

The successful detection of the translation-competent reservoir in early-treated individuals allowed us to use the STIP-Seq assay [38] to simultaneously assess the integration site and near full-length (NFL) proviral sequence from single sorted p24 + cells. In total, we sorted 156 p24 + cells from the four individuals with the highest frequencies of p24 + cells (PA02, PA14, PA34 and PA35 at UD). From these cells, we obtained (i) both the integration site and proviral genome with complete coverage in 36 p24 + cells, (ii) only the integration site in 36 p24 + cells, (iii) only the proviral genome with complete coverage in 11 p24 + cells, and (iv) no information on integration site or proviral genome with complete coverage in 73 p24 + cells (S5 Table). Integration site analysis yielded a total of 72 integration sites (S2 Table). The majority of integration sites were retrieved only once (Fig 5A), in line with the limited clonal nature of the total HIV-1 reservoir in early-treated individuals as reported in Fig 2. Clonally expanded p24 + cells were retrieved in only one out of four participants and accounted for 3% (2/72) of total integration sites. Moreover, none of the integration sites retrieved in bulk CD4 T cells and p24 + cells were overlapping (S7 Fig and S2 Table), further confirming the limited clonal nature of the reservoir shortly after treatment initiation in early-treated individuals. Proviral genome sequencing yielded a total of 47 genomes

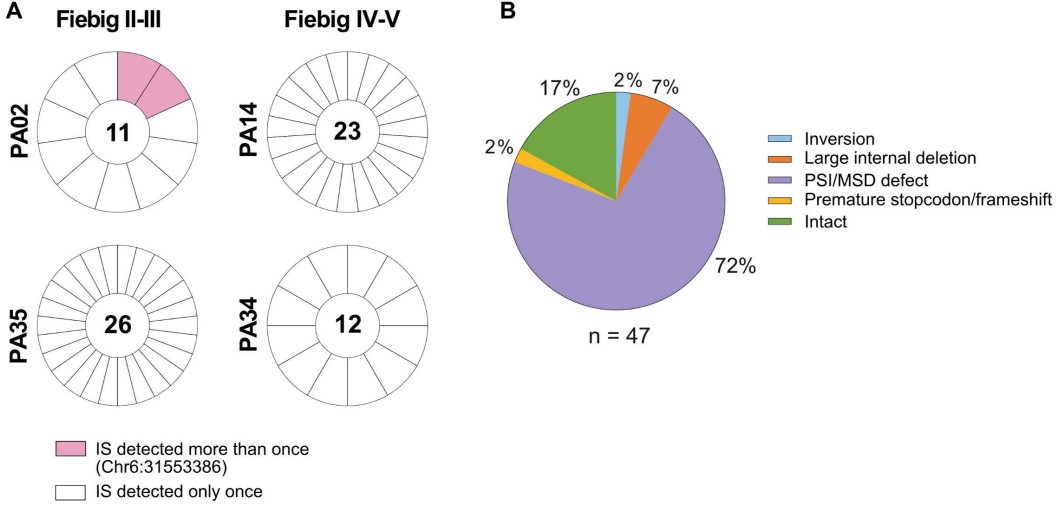

**Fig 5. Integration site analysis and near full-length (NFL) sequencing of p24 + cells.** (A) Donut charts displaying the number of HIV integration sites (IS) obtained by ISLA from p24 + cells for four participants in the corresponding Fiebig group. White parts indicate IS detected once and colored parts indicate IS detected more than once. The total number of IS generated by ISLA for each participant is shown in the middle of each donut. (B) Proportion of intact and defective proviral sequences from p24 + cells (n = 47 sequences).

with complete coverage (Figs 5B and 6). The majority of these HIV genomes displayed PSI/MSD defects (72%, 34/47), similar to what has been reported for individuals initiating ART during chronic infection [38]. In total, 8 proviral genomes were intact (17%), one provirus harbored premature stopcodon/frameshift (2%), 3 proviruses contained large internal deletions (6%, 3/47) and one provirus harbored an inversion (2%, 1/47). Proviruses with hypermutations were absent in p24 + cells. In addition, we also obtained 31 genomes with incomplete coverage for which we were only able to amplify the 3' end of the provirus (HXB2 coordinates: 5089–9602). Therefore, all complete recovered genomes were also trimmed to these coordinates to construct maximum-likelihood phylogenetic trees (Fig 6). In line with our proviral sequence dataset derived from bulk CD4 T cells (S1H Fig), the majority of STIP-Seq sequences from p24 + cells clustered together (average intra-individual genetic distance = 2 bp), confirming limited sequence diversification in early-treated individuals. Interestingly, six proviruses were located in (peri-) centromeric regions and two in genes of the ZNF family (*ZNF493* on chr19 and *ZNF484* on chr9). Notably, two of these proviruses in centromeric regions were classified as genome-intact. In conclusion, these findings indicate that stimulation with PMA/Tat-LNP enables to reactivate proviruses positioned in heterochromatin regions.

## Discussion

The mechanisms driving HIV reservoir persistence following early treatment initiation remain inadequately investigated. While prior studies have characterized the reservoir size in early-treated individuals [8–11,39], this study provides a first comprehensive longitudinal assessment of the viral reservoir in early-treated individuals, directly comparing clonal dynamics, integration site features and reservoir inducibility to chronically-treated individuals with similar treatment durations.

Our findings confirm that early initiation of ART limited the total size and phylogenetic diversity of the viral reservoir compared to ART initiation during chronic infection [7,8,11–14,40–43], increasing the chance for more effective cure strategies in early-treated individuals.

Clonal expansion is well known to contribute to the long-term persistence of the viral reservoir on ART, with studies in chronically-treated individuals showing that more than 40% of infected cells are clonally expanded after long-term ART [44–51]. In line with this, we found clonal populations of infected cells in all individuals from the Chronic UD cohort. In sharp contrast to the Chronic UD cohort, out of 452 integration sites from eight participants of the Acute UD cohort, no clonally expanded cells were detected in individuals of the Fiebig II-III group, and only three small and distinct clones of infected cells were detected in two participants of the Fiebig IV-V group. These results align with Coffin et al., who reported that clonally expanded populations of infected cells were only detected after at least one year on ART in early-treated individuals [20]. Nevertheless, longitudinal tracking of HIV integration sites beyond the first year of ART in early-treated individuals has not yet been reported. In this study we show that after five years on ART, clones were detected in three out of five participants of the Acute UD + 5 cohort, including those in the Fiebig group II-III. These findings extend previous observations by providing longitudinal evidence that clonal populations of infected cells need time to expand to a detectable size. Although clones probably exist before ART, their detection is likely obscured by the high frequency of new infection events. Once ART is initiated, these clones become detectable, and their prevalence progressively increases with time on ART. Our results confirm that clonal expansion is a key driver of reservoir persistence regardless of treatment timing.

We observed similar integration site patterns irrespective of treatment initiation timing at UD. Indeed, no significant differences were found between the acute and chronic cohorts regarding (i) the orientation of proviruses in genic regions relative to the host gene, (ii) proportions of integration sites into ZNF genes with KRAB domain on chr19 and centromeric/satellite DNA, regions associated with low transcriptional activity, (iii) integration patterns in repeat regions and Hi-C compartments, (iv) integration of proviruses in regions with lower transcriptional activity. These findings suggest that HIV proviruses persist into similar genomic regions after one year of ART, regardless of treatment initiation timing, indicating that selection mechanisms shaping the integration landscape are comparable between early and chronically-treated

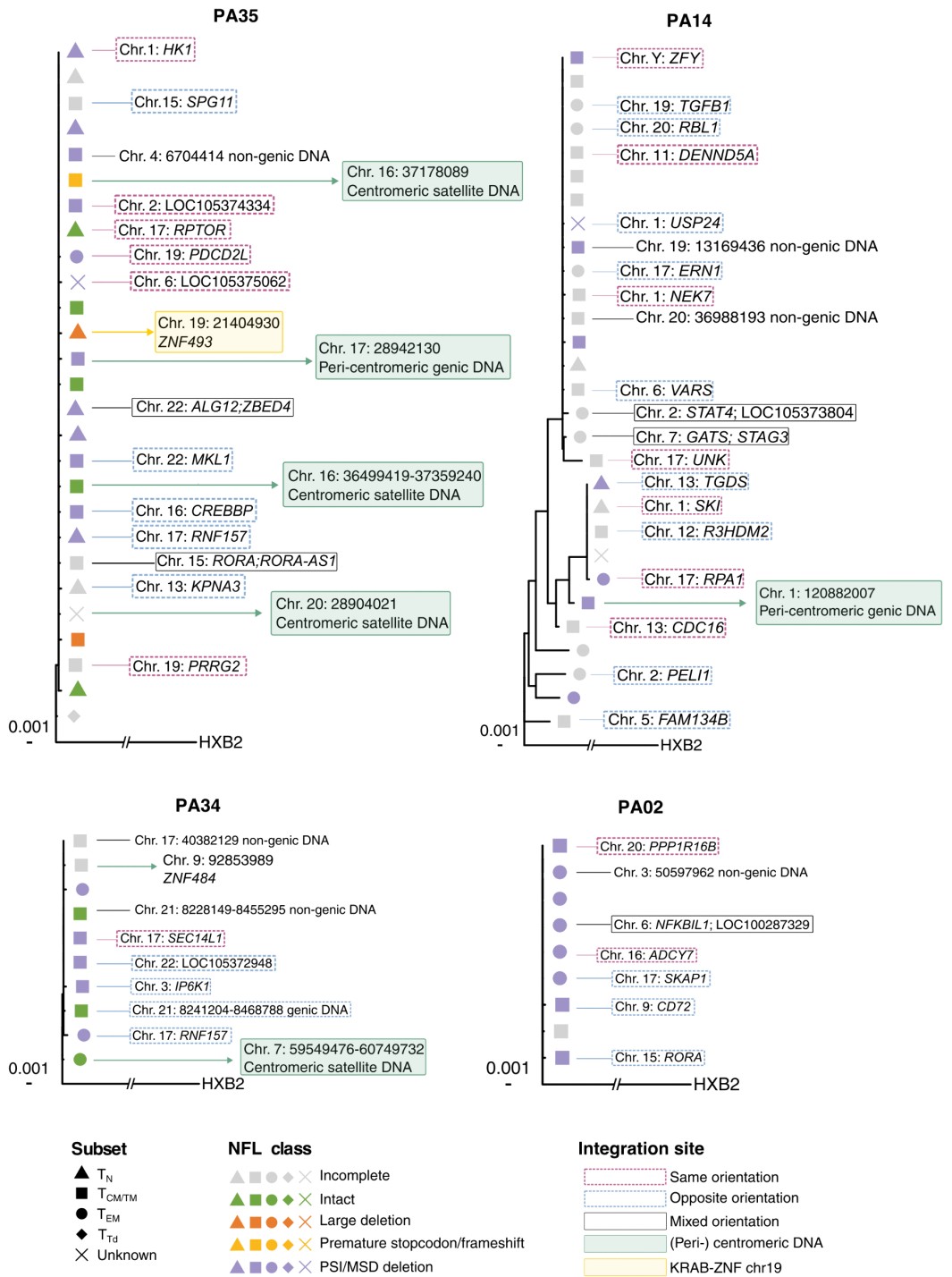

**Fig 6. Integration site analysis, near full-length proviral sequencing and phenotype analysis of p24+cells.** Maximum-likelihood phylogenetic trees for four Acute UD individuals. The trees include sequences with HXB2 coordinates 5089-9602. Naïve (TN), central/transitional memory (TCM/TTM), effector memory (TEM) and terminally differentiated (TTd) T cells are represented by triangles, rectangles, circles and diamonds, respectively. Proviral classification is color-coded. The integration sites are depicted and are color-coded according to their orientation to the host gene. (Peri-) centromeric DNA and KRAB-ZNF genes on chr19 are highlighted with a green and yellow frame, respectively. HXB2 subtype B is HIV-1 reference genome. NFL: near full-length.

individuals. However, further longitudinal studies are needed to determine whether, over extended ART duration, the better-preserved immune system in early-treated individuals exerts differential selective pressures that could alter the integration site distribution. Supporting this, a recent study in which most individuals had been on ART for more than 5 years, reported differences in integration site distribution based on ART initiation timing [52]. Furthermore, in three out of five participants of the Acute UD + 5 cohort, a selection of proviruses integrated into centromeric/satellite DNA after five years on ART was observed. Concurrently, the proportion of proviruses integrated into regions associated with higher transcriptional activity significantly decreased over time. This pattern suggests a progressive enrichment in repressive chromatin regions, mirroring the selection mechanisms observed for intact proviruses in elite controllers [22] and individuals on ART for over 20 years [26]. Our findings suggest that this selection process begins shortly after treatment initiation in early-treated individuals. A recent study investigating the dynamics of intact and defective proviruses within the first year of ART in individuals who initiated treatment during acute HIV-1 infection reported a rapid and selective enrichment of intact proviruses in heterochromatin regions [42]. However, our study extends these findings by demonstrating that this selection continues beyond the first year of ART. Although based on a small yet valuable exploratory dataset, these results suggest that post-integration selection mechanisms may shape the long-term composition of the reservoir in early-treated individuals and will need to be confirmed in larger cohorts.

Due to the small reservoir size in early-treated individuals and the lack of potent LRAs, the translation-competent reservoir remains unexplored in this cohort of individuals. In a recent study, in vitro reactivation through PMA/ionomycin stimulation did not allow to detect p24 + cells in individuals treated during acute infection under suppressive ART [21]. To allow for maximal reactivation in vitro, we used the LRAs combination PMA/Tat-LNP, which reactivates HIV at higher levels than the previously described PNB/Tat-LNP and PMA/ionomycin combination [31,53]. PMA/Tat-LNP stimulation allowed the successful detection of the inducible reservoir by HIV-Flow in 8/9 early-treated individuals on ART. Although the frequency of p24 + cells in CD4 T cells was lower in the early-treated individuals compared to the chronic group, the proportions of infected cells with an inducible provirus were similar between all cohorts, irrespective of treatment initiation timing or time on ART. This observation suggests that the challenge in detecting the translation-competent reservoir in early-treated individuals is due to a smaller inducible reservoir size rather than lower inducibility compared to individuals treated during chronic infection. These results align with prior research indicating a reduced replication-competent reservoir, as measured by the quantitative viral outgrowth assay (qVOA), in early versus late treated individuals [8]. Following stimulation, p24 + cells were mainly observed in the TCM/TTM and TEM subsets among all cohorts. However, PMA/Tat-LNP stimulation also enabled the detection of p24 + cells with a TN phenotype, confirming that this combination of LRAs successfully reactivates HIV-1 in TN infected cells. While there has been historical debate regarding the role of naïve T cells in HIV persistence, multiple recent studies have detected HIV DNA in TN cells, albeit at lower frequencies than in memory T cells [54–59]. Interestingly, after increased time on ART, we observed a shift towards a higher percentage of p24 + cells with a more differentiated phenotype, both in the acute and chronic cohorts, suggesting that the detection of reservoir cells with a naïve T cell phenotype (TN) may depend on the duration of ART. The differences in the composition of the translation-competent reservoir based on the duration of ART might be explained by the further differentiation of naïve CD4 T cells into memory phenotypes over time and/or the higher proliferative capacity of memory cells compared to TN cells. Prior research supports this explanation, by showing that TN cells can be directly infected and subsequently repopulate the memory reservoir [58,59].

We used STIP-Seq [38] to simultaneously investigate the integration site and the integrity of the proviral genome from p24 + cells from four Acute UD individuals. In agreement with our dataset generated by ISLA in bulk CD4 T cells, we observed limited clonal expansion among p24 + cells. This is in contrast with observations made in ART-suppressed individuals treated during the chronic phase of infection, where clones of p24 + cells are detected in most of these individuals [37,38]. In our study, the majority of proviruses (72%) in p24 + cells harbored PSI/MSD defects, as previously described for chronically-treated individuals [38]. These defective proviruses can contribute to chronic immune activation [60,61] and

non-suppressible viremia, but lack evidence on replication-competence [62]. 17% of the proviruses from p24 + cells were classified as genome-intact and could potentially be replication-competent, a percentage slightly higher than what was observed in individuals treated during chronic infection (9.2-12.5%) [31,38]. Interestingly, we could detect inducible intact proviruses integrated in centromeric regions, suggesting that a block-and-lock profile may be reversible. However, given that in vitro reactivation with maximal stimulation unlikely reflects reactivation in vivo, it is uncertain whether these proviruses would participate in viral rebound. Nevertheless, PMA/Tat-LNP appears to be highly effective in enabling the detection and further characterization of the translation-competent reservoir in individuals with small reservoir sizes, marking an important achievement for in vitro studies.

Our study provides important insights within a defined scope, as most participants were male and infected with HIV-1 subtype B. Due to the lack of longitudinal samples for the Chronic UD individuals, we included a cross-sectional cohort Chronic UD + x. While we performed Matched Integration site and Proviral Sequencing (MIP-Seq) for the Acute UD cohort, we present the data (IS and NFL sequences) separately, as only four intact proviruses could be linked to their respective integration site, highlighting the technical challenges of the MIP-Seq assay in retrieving simultaneously the IS and matched NFL sequence. Because the IS assay primers were subtype B-specific, only three participants from the Chronic UD cohort could be included. Additionally, because PMA significantly alters the phenotype of CD4 T cells [63], we used a restricted panel of antibodies targeting markers that are not influenced by PMA stimulation in the HIV-flow and STIP-Seq assays. Identifying additional LRAs combined with Tat-LNP that do not modify the phenotype of CD4 T cells while potently reactivating HIV will be required to further characterize the phenotypic profile of the translation-competent reservoir in early-treated individuals. Future studies with larger cohorts will be necessary to validate and extend our findings, as these are based on a relatively small sample size.

In conclusion, our study further reinforces the notion that early treatment limits clonal expansion, genetic diversity and seeding of the reservoir, making early-treated individuals a favorable group for HIV cure studies. Moreover, the comparable integration site features and inducibility of proviruses in both the acute and chronic cohorts at UD suggests that shock-and-kill strategies could be equally effective in both groups. Finally, including a longitudinal chronic cohort alongside the acute cohort in future integration site studies will be essential to determine whether the better-preserved immune system in early-treated individuals leads to an accelerated elimination of proviruses compared to individuals treated during chronic phase of infection.

## Materials and methods

### Ethics statement

All participants were provided written informed consent. Individuals from the acute cohort are part of the ACS-study, "Accurate staging of immuno-virological dynamics during acute HIV-1 infection", a multicentric prospective AHI cohort in Belgium coordinated at the HIV Cure Research Center at Ghent University (ClinicalTrials.gov identification NCT03449706). Individuals from the Chronic UD and Chronic UD + x cohort are part of the Saturne-HIV study (NCT04653610) and HIV-Mercuri (NCT04305665), respectively. All three studies and their study protocols were approved by the Ethics Committee of the Ghent University Hospital (Belgium) (EC numbers: 2017/0882; BC-08408; BC-07056).

### Participants and sample collection

Nine individuals who initiated ART during acute infection (Fiebig II-III: n = 5; Fiebig IV-V: n = 4) were included in this study (S1 Table). All participants were infected with HIV-1 subtype B. These participants were recruited at Ghent University Hospital where they underwent leukapheresis or blood sampling to collect samples of PBMCs. PBMCs were isolated by Ficoll density gradient centrifugation and were cryopreserved in liquid nitrogen. Individuals who initiated ART during chronic infection were included as reference cohorts (S1 Table).

## Negative selection of CD4 T cells

CD4 T cells were isolated from PBMCs by negative magnetic selection using the EasySep Human CD4 T Cell Enrichment Kit (StemCell Technology, #19052).

## Rainbow assay

DNA was extracted from CD4 T cells using the commercial DNeasy blood & tissue kit (Qiagen, #69504). Total and intact HIV-1 copy number were determined as previously described by the digital PCR Rainbow assay [32]. In brief, HIV-1 DNA (triplicate) and RPP30 (duplicate) reactions were run in parallel. In each reaction, 1 ug (HIV) or 5 ng (RPP30) of DNA was combined with 10 uL 4x Qiacuity probe master mix (Qiagen, #250102), 2 uL of each primer/probe set (S3 Table), 0.2 uL XbaI (NEB, #R0145M) and nuclease-free water to 40 uL total volume. Cycling was performed as follows: 2 min at 95°C; 40 cycles of (30 sec at 94°C) and 40 cycles of (1 min at 56°C). Both amplification and read-out were done on the QIAcuity Four Platform system (Qiagen, #911042). Data was analysed with the QIAcuity Software Suite (v2.1.7.182). Total HIV-1 DNA is based on RU5+partitions and intact HIV-1 DNA is based on Ψ, *env*, *gag*, *pol* quadruple-positive partitions. Intact-ness results were classified as previously described by the Rainbow assay [32]. In brief, quantifiable intactness results are obtained when at least one quadruple-positive partition is detected. If there is a signal failure in *gag* or *pol*, intactness is based on triple-positive partitions from the remaining regions. Undetectable intactness result (signal in all regions but no quadruple-positive partitions) is reported as one intact copy per total amount of input cells. We designed patient-specific primer and probes for 3 individuals (S4 Table). For this analysis, we incorporated additional data points for the Acute UD cohort beyond the aforementioned nine individuals (S1 Table). 15/38 (39%), 5/5 (100%), 4/10 (40%) and 34/57 (60%) displayed a quantifiable intactness result in the Acute UD, Acute UD+5, Chronic UD and Chronic UD+x cohort, respectively. 4/38 (10%), 1/10 (10%) and 10/57 (17%) displayed an undetectable intactness result in the Acute UD, Chronic UD and Chronic UD+x cohort, respectively. The other participants in each cohort yielded an unreportable value. Signal failure due to subtype diversity and genetic polymorphisms at primer or probe binding sites, can result in an unreportable value, a known limitation of the assay [32,64,65].

## Whole genome amplification from purified genomic DNA

DNA extracted from CD4 T cells was diluted to one HIV DNA copy per well according to total HIV-1 DNA copy number determined by the Rainbow assay. According to the Poisson distribution, if less than 30% of the reactions are positive for HIV-1 DNA, there is more than 85.7% probability that each positive reaction originates from a single HIV-1 DNA copy. Whole genome amplification was carried out on the diluted DNA by multiple displacement amplification with the REPLI-g single cell kit (Qiagen, #150345) according to the manufacturer's instructions, except that input and reagent volumes were reduced to one-fifth of the recommended amounts.

## Integration site analysis

MDA reactions were subjected to integration site sequencing by a modified version of the integration site loop amplification (ISLA) assay, as previously described [38]. For MDA reactions from genomic DNA, MDA product was diluted ½ prior to ISLA. Resulting amplicons were visualized on a 1% agarose gel and Sanger sequenced. Analysis of the sequences was performed using the 'Integration Sites' webtool (https://indra.mullins.microbiol.washington.edu/integrationsites). All ISLA primers are summarized in S3 Table.

For ISLA starting from bulk genomic DNA, digestion was performed using a maximum of 3 uL of extracted genomic DNA from CD4 T cells, 2 uL Advantage2 10x buffer (Takara Bio, #639202), 1 uL NotI (NEB, #R0189L) and nuclease-free water to a final volume of 20 uL (30 min at 37°C). Next, linear extension was performed with 10 uL of digested sample, 5 µL Advantage2 10x buffer and 0.5 µL Advantage2 polymerase (Takara Bio, #639202), 0.2 µL 10 mM dNTP, 0.75 µL 20 µM

up3.2 primer, and 33.55 μL nuclease free water. The following steps are identical to the ISLA protocol described above. Obtained integration sites are summarized in S2 Table.

### Integration site features analysis

Repetitive genomic sequences harboring HIV-1 integration sites were identified using RepeatMasker (www.repeatmasker.org) annotations from UCSC (Sep2021) for Human (hg38). ChIP-Seq data was obtained from human primary T cells from peripheral blood included in the ROADMAP database (https://egg2.wustl.edu/roadmap/web_portal/processed_data.html#ChipSeq_DNaseSeq). Centromeric regions harboring HIV-1 integration sites were identified using the UCSC human centromere annotation (https://drive.google.com/drive/folders/1sF9m8Y3eZouTZ3IEEywjs2kfHOWFBSJT?usp=sharing; hg38.UCSC.centromere). Hi-C Seq data used in this study were described by Rao et al [34], which is available in the NCBI GEO database under accession number GSE63525 (GSE63525_GM12878_subcompartments.bed.gz). It is important to note that these data were derived from the GM12878 cell line (human B-lymphoblastoids) and not from CD4 T cells, which may represent a limitation of the study. Scripts concerning these analyses can be found at the following GitHub page: https://github.com/HCRCugent/Integration_site_feature_analysis.

### Oligoclonality index

The oligoclonality index (OCI) was calculated as previously described [19,33]:

$s_i$: count of integration site $i$

$S$: Number of unique integration sites

$N = \sum_{i=1}^{S} s_i$: Total number of integration sites

$p_i = \frac{s_i}{N}$: Relative abundance of integration site $i$

$X_i = \sum_{k=1}^{i} p_k$: Cumulative abundance of all integration sites of size $\{s_i\}$ or greater

$OCI = 2 \times \left( \sum_{k=1}^{S} \left( \frac{X_k}{S} \right) - 0.5 \right)$: Oligoclonality index

### HIV-1 near-full length genome amplification

MDA wells that were ISLA positive were subjected to 5' and 3' half genome amplification. The 10 μL PCR mix for the first round is composed of: 2 μL 5X Prime STAR GXL buffer, 0.8 uL 2.5mM dNTPs, 0.2 μL PrimeStar GXL polymerase (Takara Bio, #R050B), 250nM forward and reverse primers, and 1 μL of 1/5 diluted MDA product. The mix for the second round is 25 uL PCR mix composed of: 5 μL 5X Prime STAR GXL buffer, 2 uL 2.5mM dNTPs, 0.5 μL PrimeStar GXL polymerase (Takara Bio, #R050B), 250 nM forward and reverse primers and takes 1 μL of the first-round product as an input. Thermocycling conditions for first and second PCR rounds are as follows: 2 min at 98°C; 35 cycles (10 sec at 98°C, 15 sec at 62°C, 5 min at 68°C); 7 min at 68°C. The primer sequences are summarized in S3 Table.

For STIP-Seq reactions, if the 5' reaction failed, a 2-amplicon approach (Fragment 1 and 2) for 5' was used. Briefly, 1 μL of 5' HG product from first round was added to 2.5 μL Platinum Taq High Fidelity buffer and 0.125 μL Platinum Taq High Fidelity polymerase (Invitrogen, #11304011), 1 μL 50 mM MgSO4, 0.5 μL 10 mM dNTP, 0.25 μL 20 μM forward and reverse primers, and 19.375 μL nuclease-free water. Cycling was performed as follows: 2 min at 94 °C; 35 cycles (15 sec at 94 °C, 30 sec at 55 °C, 3 min at 68 °C); 5 min at 68 °C.

FLIPS was performed on DNA extracted from total CD4 T cells with the DNeasy Blood & Tissue Kit (Qiagen, #69504), as described by Hiener et al. [55] with some minor adjustments. The assay consists of two rounds of nested PCR at an endpoint dilution where <30% of the wells are positive, using the BLOuterF and BLOuterR primers for the first round, followed by a second round using primers 275F and 280R. The first-round reaction consisted of: 1 uL genomic DNA, 1 uL Platinum Taq High Fidelity buffer and 0.05 μL Platinum Taq High Fidelity polymerase (Invitrogen, #11304011), 0.4 μL 50 mM MgSO4, 0.2 μL 10 mM dNTP, 0.1 μL 100 μM forward and reverse primers, and 7.15 μL nuclease-free water. Cycling was performed as follows: 2 min at 94°C; 3 cycles (30 sec at 94°C, 30 sec at 64°C, 10 min at 68°C); 3 cycles

(30 sec at 94°C, 30 sec at 61°C, 10 min at 68°C); 3 cycles (30 sec at 94°C, 30 sec at 58°C, 10 min at 68°C); 21 cycles (30 sec at 94°C, 30 sec at 55°C, 10 min at 68°C); 10 min at 68°C. The second-round reaction consisted of: 2 uL 1:3 diluted first round product, 2 uL Platinum Taq High Fidelity buffer and 0.1 µL Platinum Taq High Fidelity polymerase (Invitrogen, #11304011), 0.8 µL 50 mM MgSO4, 0.4µL 10 mM dNTP, 0.2 µL 100 µM forward and reverse primers, and 14.3 µL nuclease-free water. Cycling was performed as follows: 2 min at 94°C; 3 cycles (30 sec at 94°C, 30 sec at 64°C, 10 min at 68°C); 3 cycles (30 sec at 94°C, 30 sec at 61°C, 10 min at 68°C); 3 cycles (30 sec at 94°C, 30 sec at 58°C, 10 min at 68°C); 31 cycles (30 sec at 94°C, 30 sec at 55°C, 10 min at 68°C); 10 min at 68°C.

### HIV-flow procedure

Frequencies of p24-producing cells following LRA stimulation were measured by using a combination of 2 antibodies targeting the p24 protein (p24 KC57-FITC, p24 28B7-APC) as previously described by Pardons et al. [36]. The detailed protocol of the HIV-Flow procedure can be found here: dx.doi.org/10.17504/protocols.io.w4efgte. Limit of detection is reported as one p24 + cell per total number of CD4 T cells analyzed.

### Simultaneous TCR, integration site and proviral sequencing (STIP-Seq)

**CD4 T cell stimulation.** CD4 T cells were resuspended at $2 \times 10^6$ cells/mL in RPMI + 10% Fetal Bovine Serum and antiretroviral drugs were added to the culture (200 nM raltegravir, 200 nM lamivudine). Cells were rested for at least an hour at 37 °C before being stimulated with the following LRAs: 162 nM PMA (Sigma, #P8139) and Tat-LNP (250 ng/mL; 1.4 nM).

**Methanol-based HIV-Flow procedure for STIP-Seq.** The methanol-based HIV-Flow procedure was performed as previously described by Cole et al. [38]. The detailed protocol can be found here: https://protocols.io/view/methanol-based-hiv-flow-bpedmja6. In brief, following stimulation, a maximum of $5 \times 10^6$ cells per condition were resuspended in PBS and stained with fixable viability stain 510 (ThermoFisher, #L34957, 1/1000) for 20 min at RT. Cells were then stained with antibodies against cell surface molecules (CD3 AF700 Clone UCHT1 (BD Biosciences, #557943, 1/50), CD4 BV786 Clone SK3 (BD Biosciences, #563881, 1/200), CD8 BV510 Clone RPA-T8 (BioLegend, #301047, 1/200), CD45RO BV421 Clone UCHL1 (BD Biosciences, #562649, 1/100), CD27 BV605 Clone L128 (BD Biosciences, #562656, 1/100)) in PBS + 2% FBS for 20 min at 4 °C. After a 5 min-centrifugation step at 4 °C to pre-chill the cells, CD4 cells were vortexed to avoid clumping and 900 µL of ice-cold methanol (-20 °C) was gently added. Cells were fixed/permeabilized in methanol for 15 min on ice. Intracellular p24 staining was performed using a combination of 2 antibodies (p24 KC57-FITC, p24 28B7-APC) (45 min, RT) in PBS/ 1% BSA (ThermoFisher, #AM2616)/ RNAse inhibitor 0.4 U/µL (Promega, #N2615)/ DTT 1 mM. All washing steps following methanol permeabilization were done in PBS/ 0.04% BSA. In all experiments, unstimulated CD4 T cells were included to set the threshold of positivity. The gating strategy is shown in S5A Fig.

**Single cell sorting of p24 + cells.** Single p24 + cells were sorted on a BD FACSAria Fusion Cell Sorter. Cells were sorted in skirted 96-well PCR plates (Biorad, #12001925). To avoid evaporation during the sort, the PCR plate was continuously chilled at 4 °C. The cells were sorted into a volume of 2 µL PBS sc 1X (Qiagen, #150345). Index sorting, a procedure where coordinates of single-sorted cells for all markers are documented, was used to enable phenotyping of single sorted p24 + cells. CD4 T cell memory subsets were defined as follows: TN = CD45RO- CD27 + , TCM/TTM = CD45RO + CD27 + , TEM = CD45RO + CD27-, TTD = CD45RO- CD27-. Flow-Jo software v10.6.2 was used to analyze flow cytometry data (Tree-Star).

**Multiple Displacement Amplification (MDA).** Whole genome amplification of single sorted cells was carried out by multiple displacement amplification with the REPLI-g single cell kit (Qiagen, #150345), according to manufacturer's instructions, except that input and reagent volumes were halved. A positive control, consisting of 10 p24- cells sorted into the same well, was included on every plate.

**Quantitative polymerase chain reaction (qPCR) for RPP30.** After whole genome amplification by MDA, reactions were screened by qPCR on the RPP30 reference gene, as described previously [38]. Reactions that yielded a cycle of threshold (Ct) value of 38 or lower, were selected for further downstream processing.

## Proviral genome sequencing and assembly

Amplicons from near-full length genome amplification were pooled at equimolar ratios and cleaned by magnetic bead purification (Ampure XP, Beckman Coulter, #A63881), followed by quantification with the Quant-iT PicoGreen dsDNA Assay Kit (Invitrogen, #P7589). Library preparation was done with the Nextera XT DNA Library Preparation Kit (Illumina, #FC-131–1096) with indexing of 96-samples per run according to the manufacturer's instructions, except that input and reagents volumes were halved and libraries were normalized manually. The library was sequenced on a i) NextSeq Illumina platform via 2 × 150 nt paired-end sequencing with the 300 cycle v2 kit (Illumina, #MS-102–2002) according to the manufacturer's instructions, yielding approximately 200,000 reads per sample or on an ii) Element AVITI system via 2x150 nt paired-end sequencing with the Cloudbreak Freestyle kit (Element Biosciences, #860–00011). Near full-length proviral genome sequences were de novo assembled as follows: (1) FASTQ quality checks were performed with FastQC (v0.11.7, http://www.bioinformatics.babraham.ac.uk/projects/fastqc) and removal of Illumina adaptor sequences and quality-trimming of 5′ and 3′ terminal ends was performed with bbmap (v37.99, sourceforge.net/projects/bbmap/). (2) Trimmed reads were de novo assembled using MEGAHIT (v1.2.9) [66] with standard settings. (3) Resulting contigs were aligned against the HXB2 HIV-1 reference genome using blastn (v2.7.1) [67] with standard settings, and contigs that matched HXB2 were retained. (4) Trimmed reads were mapped against the de novo assembled HIV-1 contigs to generate final consensus sequences based on per-base majority consensus calling, using bbmap (v37.99, sourceforge.net/projects/bbmap/). Scripts concerning de novo assembly of HIV-1 genomes can be found at the following GitHub page: https://github.com/laulambr/virus_assembly.

## Proviral genome classification

NFL proviral genome classification was performed using the Proviral Sequence Annotation & Intactness Test (ProSeq-IT) (https://psd.cancer.gov/tools/tool_index.php) and the publicly available "Gene Cutter" and "Hypermut" webtools from the Los Alamos National Laboratory HIV sequence database (https://www.hiv.lanl.gov). Proviral genomes were classified in the following sequential order: (1) "Inversion": the presence of internal sequence inversion, defined as region of reverse complementarity. (2) "Large internal deletion": internal sequence deletion of >1000 bp. (3) "Hypermutated": APOBEC-3G/3F-induced hypermutation. For each participant, a consensus sequence was generated from the remaining genomes and a new alignment with the consensus sequence as the reference was created. This alignment was then analyzed using the 'Hypermut' webtool, with proviruses scoring $p < 0.05$ considered hypermutated. (4) "Packaging signal and/or major splice donor (PSI/MSD) defect": proviruses containing a deletion >7 bp found in any of the four stem loops of the PSI region (SL1 (HXB2: 691–734), SL2 (HXB2: 736–754), SL3 (HXB2: 766–779) and SL4 (HXB2: 790–810)). This also includes the absence and/or point-mutation of both the MSD site (sequence GT, HXB2: 744–745) and the cryptic donor site (sequence GT, HXB2: 748–749). Proviruses with a deletion covering PSI/MSD that extended into the *gag* gene, thereby removing the *gag* AUG start codon, were also classified into this category. (5) "Premature stop-codon/frameshift": premature stop-codon or frameshift caused by mutation and/or sequence insertion/deletion in the essential genes *gag*, *pol* or *env*. Proviruses with insertion/deletion >49 nt in *gag*, insertion/deletion >49 nt in *pol*, or insertion/deletion >99 nt in *env* were also classified into this category. (6) "Intact": proviruses that displayed none of the above defects were classified into this category.

## Phylogenetic analyses

Sequences were multiple aligned using MAFFT (v 7.450) [68]. Sequences with hypermutations and inversions were excluded. DIVEIN was used to calculate pairwise diversity among proviral sequences [69]. Phylogenetic trees were

constructed using PhyML v3.0 (best of NNI and 284 SPR rearrangements) and 1,000 bootstraps [70]. R (v4.2.2) [71], ggplot2 (v3.4.2) and ggtree (v3.6.2) were used for visualization and annotation of trees [72,73]. HIV-sequence data from Chronic UD + x was retrieved from GenBank OQ596859-OQ596881, OR245639-OR245784, OR245859-OR245969, OR246395-OR246586, OR246641-OR246750 and OR246848-OR246881. Phylogenetic trees from Fig 2 only include sequences with HXB2 coordinates 5089–9443.

## Statistics

Fig 3E, 3F: Two-tailed Mann-Whitney tests were performed for Acute UD-Acute UD + 5, and Acute UD-Chronic UD, for each histone feature. Figs 4A, 4B, S1A, S1C, and S1F: for pairwise comparisons between groups that did not pass normality tests, an unpaired Kruskal-Wallis test was performed, followed by a Dunn correction for multiple comparisons. Figs S1B, S1D, S1G, S6C, and S6D: for pairwise comparisons between groups that did not pass normality tests, a paired Wilcoxon test was performed. Fig 4D: Fisher's exact tests were performed between Acute UD-Acute UD + 5, and between Chronic UD-Chronic UD + x. S3 Fig: for pairwise comparisons between groups with normal residuals, a paired one-way ANOVA followed by a Tukey correction for multiple comparisons was performed. S5B Fig: Two-tailed Mann-Whitney tests were performed for Acute UD-Acute UD + 5, and Chronic UD-Chronic UD + x, for each phenotype.

## Declaration of generative AI and AI-assisted technologies in the writing process

During the preparation of this work the authors used ChatGPT in order to improve the readability of the manuscript. After using this tool/service, the authors reviewed and edited the content as needed and take full responsibility for the content of the publication.

## Supporting information

**S1 Fig. HIV reservoir quantification and phylogenetic diversity.** (A) Dot plots showing frequencies of total and intact HIV DNA copies in CD4 T cells measured by the Rainbow assay [32] for individuals of the Acute undetectable (UD, n = 38), Acute UD + 5 (n = 5), Chronic UD (n = 10), Chronic UD + x (n = 57) cohorts. Total HIV DNA is based on RU5 region, intact HIV DNA is based on Ψ, env, gag, pol regions. 15/38 (39%), 5/5 (100%), 4/10 (40%) and 34/57 (60%) displayed a quantifiable intactness result in the Acute UD, Acute UD + 5, Chronic UD and Chronic UD + x cohort, respectively. 4/38 (10%), 1/10 (10%) and 10/57 (17%) displayed an undetectable intactness result in the Acute UD, Chronic UD and Chronic UD + x cohort, respectively. Open symbols represent undetectable values. The other participants in each cohort yielded unreportable values. Grey bars depict median values. (B) Dot plots showing frequencies of total (p = 0.0625) and intact (p = 0.0625) HIV DNA copies in CD4 T cells for paired individuals from the acute cohort (n = 5). (C) Ratio of intact HIV DNA copies to total HIV DNA copies in CD4 T cells for each individual of the Acute UD, Acute UD + 5, Chronic UD and Chronic UD + x cohorts. Grey bars depict median values. Only quantifiable results are plotted. (D) Ratio of intact HIV DNA copies to total HIV DNA copies in CD4 T cells for paired individuals from the acute cohort (n = 5; p = 0.1250). (E) Spearman's rank correlation between time to ART (from estimated date of infection to treatment initiation, in days) and total HIV DNA copies in CD4 T cells from Acute UD and Chronic UD cohort. (F) Average intra-individual genetic distance for HXB2 coordinates 5089–9443. n represents the number of individuals in each cohort and each dot represents one individual. (G) Average intra-individual genetic distance for HXB2 coordinates 5089–9443 for paired individuals from the acute cohort (n = 5; p = 0.1250). (H-I) Maximum-likelihood phylogenetic trees for the acute (H) and chronic (I) cohorts. The trees include sequences with HXB2 coordinates 5089–9443. Sequences with hypermutations and inversions were excluded from this analysis. HXB2 subtype B is HIV-1 reference genome. (A,C-F) Kruskal-Wallis (Dunn correction for multiple comparisons) and (B,D,G) Wilcoxon tests were performed. Only significant p-values (p < 0.05) are represented on the graphs. (TIF)

**S2 Fig. Integration site features for the Acute UD, Acute UD + 5 and Chronic UD cohorts.** Individuals are categorized into groups accordingly to the stage when ART was initiated. Donut charts display the number of integration sites (IS) color-coded by their respective characteristic at UD and UD + 5. Distinct clonal populations of infected cells are represented by different patterns. The total number of IS generated by ISLA for each participant is shown in the middle of each donut.
(TIF)

**S3 Fig. Frequencies of p24 + cells in CD4 T cells after PMA/ionomycin, Tat-LNP/PNB and Tat-LNP/PMA stimulation.** Grey bars depict median values. Median values are plotted. Paired one-way ANOVA (Tukey's multiple comparisons test) was performed.
(TIF)

**S4 Fig. FACS analysis of p24 + cells across acute and chronic cohorts.** FACS dot plots showing the p24 KC57-FITC/28B7-APC co-staining on CD4 T cells. Red dots represent p24 + cells and frequencies of p24 + cells in CD4 T cells are depicted for each individual of the Acute UD, Acute UD + 5, Chronic UD and Chronic UD + x cohorts, accordingly to the stage of ART initiation (Fiebig II-III, Fiebig IV-V and Chronic).
(TIF)

**S5 Fig. Gating strategy and phenotypic distribution of p24 + cells for the acute and chronic cohorts.** (A) Gating strategy used in flow cytometry analysis to detect the p24 + cells. Data from PA02 is shown. (B) Frequencies of cells with a given phenotype (TN, TCM/TTM, TEM, TTd) in the p24 + fraction for Acute UD, Acute UD + 5, Chronic UD and Chronic UD + x individuals. Participants with a minimal number of 4 p24 + cells in each replicate are represented. Two-tailed Mann-Whitney tests were performed for Acute UD-Acute UD + 5, and Chronic UD-Chronic UD + x, for each phenotype. No significant p-values (p < 0.05) were obtained. Grey bars depict median values.
(TIF)

**S6 Fig. Frequencies of p24 + cells across Fiebig stages.** (A) Frequencies of p24 + cells in CD4 T cells for each individual according to the stage of ART initiation at UD, UD + 5 and UD + x. Open circles represent limit of detection (LOD). (B) Percentages of HIV-infected cells expressing the p24 protein following stimulation for each individual according to the stage of ART initiation at UD, UD + 5 and UD + x. (C) Paired longitudinal frequencies of p24 + cells in CD4 T cells for the acute cohort (p = 0.6875). Open circles represent the limit of detection (LOD). (D) Percentages of HIV-infected cells expressing the p24 protein following stimulation for paired samples of the acute cohort (p = 0.3750). (A-B) Two-tailed Mann-Whitney tests and (C-D) Wilcoxon tests were performed. Only significant p-values (p < 0.05) are represented on the graphs.
(TIF)

**S7 Fig. Comparison of integration sites from total CD4 T cells and p24 + cells.** For each participant the number of integration sites (IS) obtained by ISLA from total CD4 T cells and p24 + cells are presented. Grey parts indicate integration sites detected only once and colored parts indicate clonally expanded cells.
(TIF)

**S1 Table. Clinical characteristics of participants.** Fiebig stage at the time of diagnosis is indicated. The table shows which samples from each participant ('x') were used in the different assays. MIP-Seq ISLA refers to ISLA performed on MDA-amplified material, while ISLA refers to ISLA performed on bulk genomic DNA. VL = viral load; ART = antiretroviral therapy; UD = undetectable; MIP-Seq = Matched Integration site and Proviral Sequencing; ISLA = Integration Site Loop Amplification; HG = Half Genomes; FLIPS = Full-Length Individual Proviral Sequencing; HIV-PULSE = HIV Proviral UMI-mediated Long-read Sequencing; STIP-Seq = Simultaneous TCR, Integration site and Proviral sequencing.
(PDF)

**S2 Table. List of integration sites.** Integration sites were mapped to the GRCh38.p14 human genome reference assembly. The column Strand refers to genome orientation: strand = F indicates that HIV-1 is on the forward strand, while strand = R indicates reverse strand. The column Orientation refers to gene orientation: Same indicates that HIV-1 has the same orientation relative to the gene where it integrated, while Opposite indicates HIV-1 has the opposite orientation relative to the gene where it is integrated. Integration sites found more than once are marked by 'x' in the column Clonal. HIV-1 integrated in a ZNF KRAB gene on chromosome 19 is marked by 'x' in the column ZNF KRAB chr19. Sequence nr. = Sequence number; ISLA = Integration Site Loop Amplification; NA = not available; chr = chromosome.
(PDF)

**S3 Table. List of primers used in this manuscript.** NA = not available; ISLA = Integration Site Loop Amplification; FLIPS = Full-Length Individual Proviral Sequencing; HXB2 = subtype B HIV-1 reference genome.
(PDF)

**S4 Table. Participant-specific primer and probes.** For 3 individuals (PA35, PA02 and PC10) we made patient-specific primers and/or probe for the indicated amplicons.
(PDF)

**S5 Table. Phenotype, HIV-1 integration site and proviral sequence of p24 + cells.** Each line shows the acquired information for one sorted p24 + cell. Summary of PCR success for qPCR (RPP30 and HIV), integration site and near full-length proviral sequencing. Green color = positive PCR result, Red color = negative PCR result, Grey color = not attempted. LH = left half; RH = right half; Frag = fragment;/ = not available; N = naïve T cell; CM = central memory T cell; EM = effector memory T cell; IS = integration site.
(PDF)

**S1 Data. Source data manuscript.**
(XLSX)

## Acknowledgments

We thank all participants who donated samples to the study, as well as MDs and study nurses who helped with the recruitment and coordination of this study and the processing of blood samples. The study team thanks Maarten Verdonckt for assisting with the flow sorting experiments. We also thank NXTGNT for helping with the Illumina and AVITI sequencing. The study team also thanks Mareva Delporte for guidance of the Rainbow analysis. We are grateful for the scientific input by Sarah Gerlo.

## Author contributions

**Conceptualization:** Tine Struyve, Marion Pardons, Mathias Lichterfeld, Sofie Rutsaert, Linos Vandekerckhove.

**Data curation:** Tine Struyve, Sofie Rutsaert.

**Formal analysis:** Tine Struyve, Laurens Lambrechts.

**Funding acquisition:** Mathias Lichterfeld, Linos Vandekerckhove.

**Investigation:** Tine Struyve, Marion Pardons, Liesbet Termote, Ytse Noppe, Sofie Rutsaert.

**Methodology:** Tine Struyve, Marion Pardons.

**Project administration:** Tine Struyve, Sofie Rutsaert, Linos Vandekerckhove.

**Resources:** Jozefien De Clercq, Linos Vandekerckhove.

**Software:** Tine Struyve, Laurens Lambrechts.

**Supervision:** Marion Pardons, Sofie Rutsaert, Linos Vandekerckhove.

**Visualization:** Tine Struyve.

**Writing – original draft:** Tine Struyve.

**Writing – review & editing:** Tine Struyve, Marion Pardons, Jozefien De Clercq, Laurens Lambrechts, Mathias Lichterfeld, Sofie Rutsaert, Linos Vandekerckhove.

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
