## [Decision Letter · Decision Letter 0]

2 Oct 2025

PPATHOGENS-D-25-02083

Evolution of the HIV-1 integration site landscape and inducible reservoir in early-treated people

PLOS Pathogens

Dear Dr. Vandekerckhove,

Thank you for submitting your manuscript to PLOS Pathogens. After careful consideration, we feel that it has merit but does not fully meet PLOS Pathogens's publication criteria as it currently stands. Therefore, we invite you to submit a revised version of the manuscript that addresses the points raised during the review process, especially related to the limited sample size and the caution that should be considered in the conclusions.

Please submit your revised manuscript within 30 days Dec 01 2025 11:59PM. If you will need more time than this to complete your revisions, please reply to this message or contact the journal office at plospathogens@plos.org. Please include the following items when submitting your revised manuscript:

We look forward to receiving your revised manuscript.

Kind regards,

Mary F Kearney

Academic Editor

PLOS Pathogens

Richard Koup

Section Editor

PLOS Pathogens

Sumita Bhaduri-McIntosh

Editor-in-Chief

PLOS Pathogens

orcid.org/0000-0003-2946-9497

Michael Malim

Editor-in-Chief

PLOS Pathogens

orcid.org/0000-0002-7699-2064

**Additional Editor Comments:**

Please address the reviewers comments especially related to the limited sample size and conclusions.

**Journal Requirements:**

At this stage, the following Authors/Authors require contributions: Tine Struyve, Marion Pardons, Jozefien De Clercq, Liesbet Termote, Laurens Lambrechts, Ytse Noppe, Mathias Lichterfeld, Sofie Rutsaert, and Linos Vandekerckhove. Please ensure that the full contributions of each author are acknowledged in the "Add/Edit/Remove Authors" section of our submission form.

5) When completing the data availability statement of the submission form, you indicated that you will make your data available on acceptance. We strongly recommend all authors decide on a data sharing plan before acceptance, as the process can be lengthy and hold up publication timelines. Please note that, though access restrictions are acceptable now, your entire data will need to be made freely accessible if your manuscript is accepted for publication. This policy applies to all data except where public deposition would breach compliance with the protocol approved by your research ethics board. If you are unable to adhere to our open data policy, please kindly revise your statement to explain your reasoning and we will seek the editor's input on an exemption. Please be assured that, once you have provided your new statement, the assessment of your exemption will not hold up the peer review process.

6) Your current Financial Disclosure states, "Yes ↳ Please add funding details. This current research work was supported by the NIH (R01 AI152979-01, MPI: L.V. and M.L.), NIH MDC grant (RID-HIV: UM1AI164561) and by fund for scientific research (FWO JUNIOR: G0B3820N, FWO SENIOR: G0A1824N and G0ABH25N), SBO-SAPHIR (S000319N). The content is solely the responsibility of the authors and does not necessarily represent the official views of the National Institutes of Health. T.S. was supported by FWO Vlaanderen (1SA7125N). L.V. was supported by the Research Foundation Flanders (1.8.020.09.N.00). M.P. was supported by postdoctoral funding from VLAIO O&O (HBC.2018.2278). J.DC. is a beneficiary of a Ghent University Special Research Fund (BOF) doctoral scholarship (BOF19/DOC/220). L.L was supported by FWO Vlaanderen (1S29220N). Work in the lab of M.L. is supported by NIH grants AI184094, AI152979, AI155233, AI135940, AI176579. M.L. is a member of the DARE, ERASE, PAVE and BEAT-HIV Martin Delaney Collaboratories (UM1 AI164560, AI164562, AI164566, AI164570). The funders had no role in study design, data collection and analysis, decision to publish, or preparation of the manuscript. ↳ Please select the country of your main research funder (please select carefully as in some cases this is used in fee calculation). BELGIUM - BE".

However, your funding information on the submission form indicates an alternative arrangement for the funders. 

Please indicate by return email the full and correct funding information for your study and confirm the order in which funding contributions should appear. Please be sure to indicate whether the funders played any role in the study design, data collection and analysis, decision to publish, or preparation of the manuscript.

**Reviewers' Comments:**

Reviewer's Responses to Questions

**Part I - Summary**

Reviewer #1: This longitudinal study compares characteristics of HIV-infected cells collected from participants who initiated antiretroviral treatment (ART) during acute or chronic stages of infection, respectively, with sampling shortly after participants' viremia became suppresseed to undetectable levels and several years thereafter. Infected cells were characterized in several aspects, including frequencies, proviral intactness, clonality and clonal dynamics, differentiation state, integration sites and chromatin environments thereof, and inducibility using potent latency reversal agents (tat mRNA, PMA).

Among several notable findings, the authors report that although demonstrated clonal populations of infected cells are generally less abundant in participants treated in acute infection, they are established early and the fraction of demonstrated expanded clones increases with time on ART. The fraction of infected cells with proviruses integrated into centromeric/satellite regions also increases over time in this cohort. The capacity to induce some of these infected cells into expressing p24 by treatment with potent LRAs was likewise a hopefully and interesting finding. Even with these highlights, data analysis was balanced overall and all findings, including some where no significant diffferences between experimental populations were observed, were duly reported.

Reviewer #2: This manuscript by Struyve et al. presents a longitudinal analysis of the HIV-1 reservoir in individuals initiating ART during acute infection, compared to chronically treated cohorts. Using cutting-edge techniques—integration site loop amplification, the Rainbow assay, STIP-seq and HIV-Flow with Tat-LNP stimulation—the authors evaluate reservoir size, clonal dynamics, integration site distributions, and the inducible reservoir. The main findings of the study are that clonal expansion becomes increasingly detectable over time in early-treated individuals, proviral integrations in heterochromatin regions are present even after one year on ART and become enriched after five years, Tat-LNP in combination with PMA allows detection of inducible reservoirs in early-treated individuals with very small reservoir sizes, and the inducible reservoir shifts toward more differentiated T cell subsets with increasing time on ART. The work is technically rigorous and advances our field, particularly in its integration of multiple orthogonal assays. However, several claims are overstated relative to the dataset, and important limitations should be more clearly acknowledged.

Reviewer #3: The authors present a detailed and thorough analysis of the HIV-1 reservoir, focusing on longitudinal analysis in individuals initiating ART during acute infection. Major findings focus on integration site landscape and dynamics, relative infection rate of CD4 T-cell subsets, and latency reversal/reactivity with Tat-LNP and PMA. The authors present thorough multi-parametric analysis and the raw/supporting data and methods detail provided is, generally, quite extensive. As acknowledged, sampling, numbers is integration sites/proviral sequences, and number of participants (gender, HIV clade) is a limitation that this study and most in the field face. The authors stress the novelty of acute (ART) in combination with longitudinal reservoir characterization, and findings here may support more extensive (longitudinal follow up, additional donors) future investigation of this topic. This review does not feel experimental revision/follow-up is needed; however, would like to call attention to some data analysis/presentation that may clarify/strengthen the authors conclusions and message.

**Part II – Major Issues: Key Experiments Required for Acceptance**

Reviewer #1: None.

Reviewer #2: 1) The acute cohort includes nine individuals, with only five followed longitudinally to UD+5. The chronic cohorts are cross-sectional, and only three chronic participants were included for integration site analysis due to primer bias. These limitations restrict the ability to draw strong conclusions about differences between acute and chronic initiation or about longitudinal trends. The authors should temper generalizations and clearly frame this as a small but valuable exploratory dataset.

As an example:

The results section states that “integration into heterochromatin regions is already detectable early after ART initiation in the acute cohort and significantly increases over time on ART.” While centromeric or satellite integrations were seen, the absolute numbers are small and enrichment is driven by single individuals.

A more accurate representation would be that "such integrations are detectable in some acute participants and enriched in certain individuals over time, consistent with the possibility of early selection for proviruses in heterochromatin."

2) The finding that clonal expansion increases over time in early-treated individuals is not unexpected and has been described previously, for example in Coffin et al., JCI Insight 2019. The novelty here lies in the longitudinal confirmation within an acute cohort, rather than in demonstrating the phenomenon itself. The authors should adjust their framing accordingly.

3) The observation of increasing integration in centromeric or satellite DNA and KRAB-ZNF genes at UD+5 is intriguing, but the absolute numbers are small, enrichment is sometimes driven by single participants. The conclusions about post-integration selection should therefore be presented more cautiously. The use of Tat-LNP with PMA is an important technical strength and clearly improves detection sensitivity, but this is an extreme stimulation condition, and it is unclear whether proviruses detected under these conditions—especially those integrated in heterochromatin—would be inducible in vivo or relevant for rebound. The authors should add stronger caveats re: this point.

Reviewer #3: Though not experimental, this reviewer would like to address data analysis questions in key figures.

1. Figure 2, integration site clones: Figure 2 focuses focus on the bulk IS landscape of UD, UD+5 vs control. Limited sampling at some conditions (7, 20, 26) limits the ability to capture a complete picture of clonality; however, remarkably, the detection of clones with intra/inter-sample, does speak to an apparently large clone size such that they may be detected at limited sampling. Rather than or in addition to percent clone/total IS, a calculation of oligoclonality or clonal diversity many allow for a quantitative comparison across acute, acute+5, and control.

2. Figure 6, phylogenetic diversity: The authors note a lack of phylogenetic diversity in acute (acute+5) proviral sequences relative to chronic. This is consistent with the field and supported for acute in F6. Is there any quantification for diversity from the chronic cohort, and comparing something like average pairwise distance (APD) between the groups? Such analysis quantification would lend support to “Our findings confirm that early initiation of ART limited the total size and phylogenetic diversity 270 of the viral reservoir compared to ART initiation during chronic infection.”

3. Figure 4: The subset conclusion is supported by Figure 4C; however, 4D at current does not seem to contribute to conclusions, comparing %p24 +/- by Fisher’s exact, which shows more p24+ in EM and less in Naïve. Suggest showing 4D longitudinal, ie %p24+ Acute vs Acute+5, Chronic vs Chronic+x. Can be done non-paired due to missing time points.

**Part III – Minor Issues: Editorial and Data Presentation Modifications**

Reviewer #1: The paper is well written, the figures clear and of high quality, and the analysis is insightful yet balanced. My principal yet minor concern is that, in multiple instances, it is unclear whether the authors consider integration into peri-centromeric genic DNA, centromeric satellite DNA, "KRAB-ZNF genes on chr19", and KRAB-ZNF genes elsewhere to be equivalent in terms of heterochromatin environment and/or as regions of generalized transcriptional suppression as a basis for enabling persistence of proximally integrated intact proviruses. Some clarification and/or use of more precise language in these instances would be welcome.

Reviewer #2: - Many individuals yielded unreportable results with the Rainbow assay (Supplementary Fig. 1). The limitations of the assay in this context should be discussed.

- The manuscript consistently refers to the “translation-competent reservoir.” For clarity, the authors should define precisely what Tat-LNP HIV-Flow measures and distinguish it from replication-competent virus.

Reviewer #3: 1. Figure 3, the text says KRAB-ZNF chr19, but figure text says KRAB-ZNF (total?), please clarify.

2. Language Note: Take care to distinguish/acknowledge the impact between ‘preferentially integrated’ (which may have high impact at short timepoint) vs selected/persisted (longer timepoint) (see below line numbers)

a. “Analysis of integration in repetitive elements 164 using RepeatMasker showed that proviruses preferentially integrated into long and short 165 interspersed nuclear elements (LINE, SINE) among all cohorts (Acute UD: 41%; Acute UD+5: 166 31%; Chronic UD: 43% of total integration sites; Figure 3D).”

b. “Furthermore, three out of five participants of the Acute UD+5 cohort showed increased 305 integration”

3. Authors sometimes ‘lump’ KRAB-ZNF and centromere/satellite together in text and thematically. Care should be taken to parse the two groups (genic vs non-genic, chromosomal location, etc) and distinguished as such.

a. Figure 6, grouped together in solid green box

b. Figure 2 and text lines 149-153, “Notably, proviral integration in centromeric/satellite DNA (0.88%150 of total IS at UD) and KRAB-ZNF genes on chr19 (0.88% of total IS at UD) was already151 detectable in the acute phase of infection (Figure 3B). Moreover, the acute and chronic cohorts 152 showed similar levels of integration within these regions at UD (Acute UD: 1.77% and Chronic153 UD: 3.55%; p = 0.17).”

i. “These regions” unclear KRAB-ZNF or cent/sat or both?

4. Language Note: Conclusion casts a wide net with catch-all “heterochromatin” rather parameters defined in data.

a. “Interestingly, integration into heterochromatin 183 regions is already detectable early after ART initiation in the acute cohort and significantly 184 increases over time on ART.

5. Proviral Sequence/Question: It is of note that STIP-SEQ p24+ proviral/IS sequencing resulting in only 17% intact provirus observed. p24 detection in non-replication competent provirus is of note for 2 points (1) defective provirus protein expression and immune activation and (2) LRA studies assume all p24+ represent reservoir/viremia recrudescent provirus, as such, may overestimate underlying reservoir and/or impact of LRA on true reservoir target. Please make complete/incomplete proviral genomes publicly available.

6. Citation of pediatric cohort/reservoir characterization. May note in introduction previous studies that have characterized integration site landscape in pediatric cohorts (cross-section) as ‘early/acute,’ though not Fiebig as done here.

a. Bale Kearney ISA Children mBio 2021

b. Garcia-Broncano, Kuritzkes, Lichterfeld STM Neonates STM 2019

PLOS authors have the option to publish the peer review history of their article (what does this mean? ). If published, this will include your full peer review and any attached files.

**Do you want your identity to be public for this peer review?** For information about this choice, including consent withdrawal, please see our Privacy Policy .

Reviewer #1: No

Reviewer #2: No

Reviewer #3: No

**Figure resubmission:**
---

## [Editor Report · Decision Letter 1]

7 Nov 2025

Dear Prof. Dr. Vandekerckhove,

We are pleased to inform you that your manuscript 'Evolution of the HIV-1 integration site landscape and inducible reservoir in early-treated people' has been provisionally accepted for publication in PLOS Pathogens.

Best regards,

Richard A. Koup, M.D.

Section Editor

PLOS Pathogens

Richard Koup

Section Editor

PLOS Pathogens

Sumita Bhaduri-McIntosh

Editor-in-Chief

PLOS Pathogens

orcid.org/0000-0003-2946-9497

Michael Malim

Editor-in-Chief

PLOS Pathogens

orcid.org/0000-0002-7699-2064
---

## [Editor Report · Acceptance letter]

Dear Prof. Dr. Vandekerckhove,

We are delighted to inform you that your manuscript, "Evolution of the HIV-1 integration site landscape and inducible reservoir in early-treated people," has been formally accepted for publication in PLOS Pathogens.

Best regards,

Sumita Bhaduri-McIntosh

Editor-in-Chief

PLOS Pathogens

orcid.org/0000-0003-2946-9497

Michael Malim

Editor-in-Chief

PLOS Pathogens

orcid.org/0000-0002-7699-2064